# Model Extraction Attacks on Split Federated Learning

## Abstract

Federated learning (FL) is a popular collaborative learning scheme involving multiple clients and a server. FL focuses on client's data privacy but exposes interfaces for Model Extraction (ME) attacks. As FL periodically collects and shares model parameters, a malicious client can download the latest model and thus steal model Intellectual Property (IP). Split Federated Learning (SFL), a recent variant of FL, splits the model into two, giving one part of the model (client-side model) to clients, and the remaining part (server-side model) to the server. While SFL was primarily designed to facilitate training on resource-constrained devices, it prevents some ME attacks by blocking prediction queries. In this work, we expose the vulnerability of SFL and show how ME attacks can be launched by malicious clients querying the gradient information from server-side. We propose five ME attacks that differ in the gradient usage in data crafting, generating, gradient matching and soft-label crafting as well as in the attacker data availability assumptions. We show that the proposed ME attacks work exceptionally well for SFL. For instance, when the server-side model has five layers, our proposed ME attack can achieve over 90% accuracy with less than 2% accuracy degradation with VGG-11 on CIFAR-10.

## 1 Introduction

Federated Learning (FL) [McMahan et al., 2017] has become increasingly popular thanks to its ability to protect users' data privacy and comply with General Data Protection Regulation (GDPR) policy. In FL, clients locally update their model copies, and the FL server collects them by averaging the model parameters, then distributing the averaged model again to its clients. Such a setting only allows model parameters to be shared with the server, and direct data sharing is avoided. One drawback of FL is its clients need to train the entire model locally, which is usually challenging for resource-limited edge devices. As the countermeasure, Split Federated Learning (SFL) scheme [Thapa et al., 2020] is proposed as a variant of FL. In SFL, a neural network is split into a client-side model and a server-side model, where the client-side model is shared among multiple clients and processed locally on their devices. During training, clients offload the intermediate activations to server, where the heavy-duty computation is performed at the heavy-duty server and the computed gradients are sent back to clients. SFL follows the same model averaging routine as FL to synchronize the model. SFL avoids collecting clients' raw data and also reduces the computational overhead at the client-end.

In addition to the computation advantage introduced by SFL, it can also provide model IP protection which is absent in FL. The high training cost of high-performance NN makes the NN model a valuable Intellectual Property (IP). Unlike FL which handles the entire NN over to the clients, SFL preserves the server-side model which prevents potential IP theft (Fig. 1 (a)). Moreover, according to our investigation, SFL shows resistance to Model Extraction (ME) attack [Tramèr et al., 2016, Jagielski et al., 2020]. In an ME attack, the model IP can be acquired by querying a publicly accessible

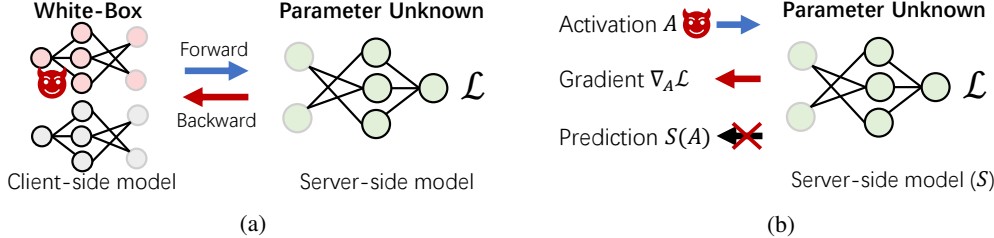

Figure 1: Model Extraction (ME) attack in SFL. (a) The attacker knows the client-side model parameters but does not know the server-side model parameters. (b) Server-side model allows gradient query access but does not allow prediction queries.

prediction API of the model. Prior ME attacks cannot be applied to SFL as the protocol does not allow prediction query access, as illustrated in Fig. 1 (b).

As prior ME attacks fail in SFL, we propose SFL-specific ME attacks in this work. Assuming the client-side model is a white box, the attacker posing as a participant client can get gradient information of inputs sent to the server. We propose five variants of ME attacks on SFL enumerated as Craft-ME, GAN-ME, GM-ME, Train-ME, and SoftTrain-ME. These ME attack variants extensively cover different gradient usage, including data crafting, data generating, gradient matching and soft label crafting, as well as assumptions on data such as no data (including randomly generated noise data) [Truong et al., 2021], only auxiliary data (out-of-distribution data) and training data (in-distribution data). We consider both train-from-scratch and fine-tuning SFL applications as gradient consistency is different. We benchmark the performance of five ME attacks on SFL for these two cases and show that ME attacks can succeed without any data in fine-tuning applications. We also show that ME performance strongly correlates with the #layers in the server-side model and fail when the #layers in the server-side model increase. However, increasing layers in server-side model reduces layers in client-side model, which compromises clients' privacy. Finally, we conclude that using L1 regularization during training can improve SFL schemes' resistance to ME attacks. In summary, we make the following contributions:

- We define the unique threat model in SFL and propose five ME attacks. These attacks differ in the usage of gradients (data crafting, data generating, gradient matching and soft label crafting) and data assumptions (no data, only auxiliary data and training data). To our knowledge, this is the first work that studies ME attacks for SFL.

- We study the performance of the proposed attacks and find that even when the attacker has no data, the model can be extracted with high accuracy. With auxiliary data or with training data, ME attack performance can be further improved. For a 5-layer-in-server SFL, the strongest ME attack can derive a surrogate model with over 90% accuracy, and less than 2% accuracy degradation compared to the original VGG-11 on CIFAR-10 model.

- We find the ME attack performance decreases when more layers are present in server-side model. However, such a split model configuration compromises clients' privacy. We also show resistance to ME attack can be improved by regularizing the client-side model.

## 2 Related Work

**Model-split learning schemes.** The key idea for model-split learning schemes is to split the model so that part of it is processed in the client and the rest is offloaded to the server. This idea was first proposed in Kang et al. [2017], Teerapittayanon et al. [2017], Liu et al. [2018] for inference tasks. Gupta and Raskar [2018] then extended this idea for split learning, a collaborative multi-client neural network training. However, the round-robin design needed clients to learn sequentially and thus has a huge disadvantage in terms of training time.

**Split federated learning (SFL).** In this paper, we consider the SFL scheme where the clients process their local models in parallel followed by periodic synchronization as in FedAvg [McMahan et al., 2017]. This is the SFL-V2 scheme that is introduced in Thapa et al. [2020] for its better accuracy performance. The detailed process is shown in Algorithm 1. At the beginning of each epoch, server performs the synchronization of client-side model and sends the updated version to all clients. Then,

---

**Algorithm 1** Split Federated Learning

---

**Require:** For $M$ clients, instantiate private training data $(\mathbf{X}_i, \mathbf{Y}_i)$ for $1, 2, ..., M$. Server-side model $S$ has $N$ layers and client-side model $C_i$ has $L - N$ layers.

1: initialize $C_i, S$
2: **for** epoch $t \leftarrow 1$ to num_epochs **do**
3:     $C^* = \frac{1}{M} \sum_{i=1}^{M} C_i$                         {Model Synchronization}
4:     $C_i \leftarrow C^*$ for all $i$
5:     **for** step $s \leftarrow 1$ to num_batches **do**
6:         **for** client $i \leftarrow 1$ to $M$ **in Parallel do**
7:             data batch $(\boldsymbol{x}_i, \boldsymbol{y}_i) \leftarrow (\mathbf{X}_i, \mathbf{Y}_i)$
8:             $\boldsymbol{A}_i = C_i(\boldsymbol{W}_{C_i}; \boldsymbol{x}_i)$                    {Client forward; send $\boldsymbol{A}_i$ to Server}
9:         **end for**

---

10:         **for** client $i \leftarrow 1$ to $M$ **in Sequential do**
11:             $\mathcal{L} = \mathcal{L}_{CE}(S(\boldsymbol{W}_S; \boldsymbol{A}_i), \boldsymbol{y}_i)$                         {Server forward}
12:             $\nabla_{\boldsymbol{A}_i} \mathcal{L} \leftarrow$ back-propagation              {Server backward, send $\nabla_{\boldsymbol{A}_i} \mathcal{L}$ to Client}
13:             Update $\boldsymbol{W}_S$;
14:         **end for**

---

15:         **for** client $i \leftarrow 1$ to $M$ **in Parallel do**
16:             $\nabla_{\boldsymbol{x}_i} \mathcal{L} \leftarrow$ back-propagation                         {Client backward}
17:             Update $\boldsymbol{W}_{C_i}$;
18:         **end for**
19:     **end for**
20: **end for**

---

clients perform forward propagation locally till layer $L - N$ (the last layer of client-side model), sending the intermediate activation $\boldsymbol{A}_i$ to the server (line 8). Server accepts the activation and label $\boldsymbol{y}_i$ sent from clients, and uses them to calculate the loss and initiates the backward process (line 9). The backward process (line 10) consists of several steps: server performs backward propagation on the loss, updates server-side model and sends back gradient $\nabla_{\boldsymbol{A}_i} \mathcal{L}$ to clients. Clients then continue the backward propagation on their client-side model copies and perform model updates accordingly. While FedGKT [He et al., 2020] leaks prediction logits to clients, SFL does not, making it a promising candidate against ME attacks.

**Model extraction (ME) attack.** ME attack targets model prediction service APIs and retrieves confidence score or prediction label for given inputs. The vulnerability of a DNN model to ME is first shown in Tramèr et al. [2016]. Jagielski et al. [2020] shows that high fidelity model extraction can be achieved with fewer queries and the surrogate model can be used for launching more successful adversarial attacks. A successful ME attack not only breaches the model IP, but also makes the model more vulnerable to attacks. ME attack also supports transferable adversarial attacks [Goodfellow et al., 2014], mainly targeted ones [Madry et al., 2017] against the victim model. It can also be used to perform bit-flip attacks [Rakin et al., 2019]; with hardware expertise, a few bit flips on model parameters can degrade ResNet-18 model accuracy to below 1%.

**Data privacy in SFL.** Similar to FL, SFL scheme also has data privacy concerns. The most serious one is its vulnerability to MI attacks. In model-based MI attack [Fredrikson et al., 2015], the attacker trains an inverted version of client-side model and can directly reconstruct raw inputs from the intermediate activation. Recent works [Vepakomma et al., 2020, Li et al., 2022] point out this vulnerability and provide practical ways to mitigate MI. However, defenses only work well if the client-side model has enough number of layers.

## 3   Threat Model

### 3.1   Attacker assumptions.

**Objectives.** According to Jagielski et al. [2020], there are three model extraction (ME) attack objectives: i) functional equivalence, ii) high accuracy, and iii) high fidelity. However, achieving

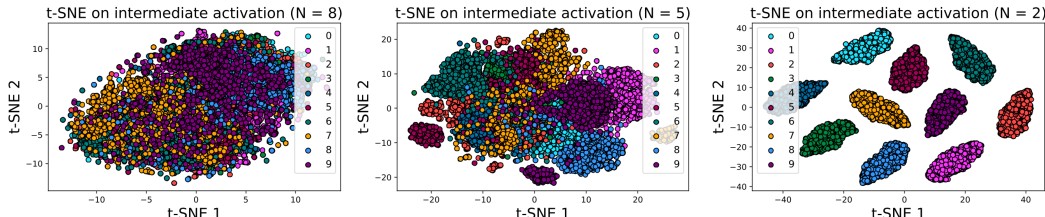

Figure 2: tSNE analysis of intermediate activation with different $N$ (8, 5, 2 from left to right) on a VGG11 model on CIFAR-10 dataset. $N$ denotes number of layers in server-side model in SFL.

functional equivalence is difficult in practical applications. Hence, most of the existing practical ME attacks on NN models focus on achieving high accuracy and fidelity. To achieve the accuracy goal, the attacker wants to obtain a model that maximizes the prediction correctness. To achieve the fidelity goal, the attacker wants to derive a model with a similar decision boundary as the victim model before launching adversarial attacks [Biggio et al., 2013].

**Capabilities.** We assume the attacker acts as a client in a multi-client SFL scheme against the server (model owner). Consider the SFL scheme outlined in Fig. 1. We assume the entire model has a total of $L$ layers (or layer-like blocks, i.e. BasicBlock in ResNet) out of which the server processes $N$ layers. The attacker holds *white-box assumption* on the client-side model (consists of $L - N$ layers), that is, it knows the exact model architecture and parameters for those layers. The attacker holds a *grey-box assumption* on the $N$-layer server-side model, that is, it knows its architecture and loss function while the model parameters are unknown. Also, we assume *server blocks the prediction queries* thus neither logits nor prediction labels are accessible by clients during training, but server *allows gradient queries* to let client-side models be updated. Based on a client's activation $\boldsymbol{A} = C(\boldsymbol{x})$ and its label $\boldsymbol{y}$, gradient information $\nabla_{\boldsymbol{A}} \mathcal{L}$ is computed and sent back to clients. In addition, we assume the attacker can perform gradient queries on any input $\boldsymbol{x}$, including malicious ones.

### 3.2 Analysis

**Partial model extraction problem.** Since the attacker already has a white-box assumption of the client-side model, the attacker only needs to extract the server-side model to reveal the entire model. This results in an easier problem setting. We observe client-side model heavily regularizes the feature space of its output (input of server-side model), making ME attacks easier to succeed, especially when $N$ is small. As shown in Fig. 2, on a VGG-11 [Simonyan and Zisserman, 2014] model on CIFAR-10 dataset, as $N$ becomes smaller, tSNE embeddings of intermediate features with different labels are easier to distinguish. For $N = 2$, ME attack is as simple as separating different clusters with a linear layer.

To show that the extracting part of the model with ME is easier, we study existing ME attacks on the server-side model, **by assuming that prediction access is allowed**. Specifically, we investigate CopyCat CNN [Correia-Silva et al., 2018], Knockoff-random [Orekondy et al., 2019] and data-free ME [Truong et al., 2021]. As shown in Fig. 3 (a), with auxiliary data (CIFAR-100) and enough query budget, both attacks derive a surrogate model with very high accuracy even for a large $N$ setting. Moreover, attacker with no data can also succeed with data-free ME as shown in Fig. 3 (a). When the query budget is equal to 2 million, the data-free ME can extract the model with high accuracy even when $N$ is equal to 5.

**Consistency of gradient query.** For fine-tuning applications [Park et al., 2021], attackers get consistent gradient information from gradient query, as server-side model parameters are frozen or updated with a very small learning rate. Gradient consistency is very beneficial for ME attacks. However, for a training-from-scratch usage, queries to SFL model obtain inconsistent gradient information as the server-side model drastically changes during training. As shown in Fig. 3 (c), for the same query input, the gradient is drastically different in different epochs.

## 4 Proposed Model Extraction Attack

Previously, Milli et al. [2019] demonstrated that using gradients to reveal one-layer linear transformation is trivial. Given $f(\boldsymbol{x}) = \boldsymbol{W}^T \boldsymbol{x}$, one can directly infer $\boldsymbol{W}$ from a single gradient query given that

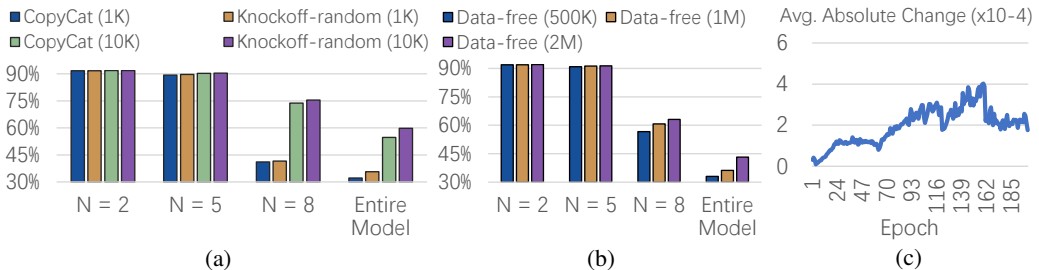

(a)                        (b)                        (c)

Figure 3: Analyses of SFL threat model. Existing ME attack performance on VGG-11 CIFAR-10 model with different $N$, assuming prediction query access is allowed: (a) ME attacks using CIFAR-100 as auxiliary dataset; (b) Data-free ME attack that demonstrates ME attack on part of the model is much easier than ME attack on the entire model; (c) Inconsistent gradient problem in training-from-scratch SFL. The y-axis denotes the change in gradient (lower means more consistent) for the same inputs in different epochs.

$\boldsymbol{W}^T = \nabla_{\boldsymbol{x}} f(\boldsymbol{x})$. However, using gradient only can go no further than one layer. Milli et al. [2019] shows that to recover a two-layer ReLU network of the form $f(\boldsymbol{x}) = \sum_{n=1}^{h} g(\boldsymbol{x})_i \boldsymbol{W}_i \boldsymbol{A}_i^T \boldsymbol{x}$, where $g(\boldsymbol{x}) = \mathbb{1}\{\boldsymbol{A}\boldsymbol{x} > 0\}$, $\boldsymbol{A}$ is of $\mathbb{R}^{h \times d}$ and $\boldsymbol{W}$ is of $\mathbb{R}^h$, using input gradient can recover the absolute value of normal vectors $|\boldsymbol{W}_i \boldsymbol{A}_i|$ for $i \in [h]$. In order to get the sign information of $\boldsymbol{W}_i \boldsymbol{A}_i$, prediction query is required which is not supported by SFL's threat model.

So in this paper, we investigate approximate ME attacks that differ in the data assumptions and gradient usage. We propose five ME attacks as shown in Table 1. We also include a naive baseline scheme that train the surrogate model from scratch without access to either the client-side model or gradients. Despite the differences, the five proposed attack methods follow the same strategy, that is, they all train a randomly initialized surrogate server-side model from scratch. We provide a detailed illustration of proposed five attacks in Appendix A.2.

Table 1: Model Extraction Attack Methods in SFL

| Method | Data Assumption | Prediction Query | Gradient Usage | Client-side model Usage |
|---|---|---|---|---|
| Craft-ME | None | No | Data Crafting | Initialization |
| GAN-ME | None | No | Data Generator | Initialization |
| GM-ME | Natural Auxiliary | No | Gradient Matching | Initialization |
| Train-ME | Limited Training | No | None | Initialization |
| SoftTrain-ME | Limited Training | No | Soft Label Crafting | Initialization |
| Naive Baseline | Limited Training | No | None | None |

### 4.1 ME attacks without training data

We first consider two cases where the attacker does not have training data: one is that attacker does not use any data or use randomly generated noise data. This case is motivated by Truong et al. [2021] which shows that when an attacker does not have a *similar enough* dataset, it is better not to use it at all. The other case is when the attacker has an auxiliary dataset with different labels from the victim's training data (for example, CIFAR-10 and CIFAR-100).

**Crafting model extraction (Craft-ME).** Inspired by Han et al. [2018], where data-label pairs (referred to as instances) with small-loss are shown to present useful guidance for knowledge distillation, we propose a simple method to craft small-loss instances using gradient queries and use them to train the surrogate model. We initialize random input $\boldsymbol{x}_r$ for every class label $c$, and use the gradient $\nabla_{\boldsymbol{x}_r} \mathcal{L}$ to update $\boldsymbol{x}_r$. For each input, updating is repeated for a number of steps. By varying label $c$, a collection of small-loss instances is derived during SFL training. Then, a surrogate model is trained from scratch on these small-loss instances.

**GAN-based model extraction (GAN-ME).** Recent work [Truong et al., 2021] proposes a GAN-based approach for data-free ME. The key idea is to use a generator $G$ to continually feed fake inputs to the victim model $V$ and surrogate model $S$, and use confidence score matching to let $S$ approach $V$. However, the confidence score matching needs prediction query which is not allowed in our case.

Thus, we adapt the GAN-based method to gradient-query-only case and propose a two-step method: First, a conditional-GAN (c-GAN) model $G(z|c)$ is initialized. The attacker trains the generator during victim model's training by generating fake data $\boldsymbol{x}_f$ and label $c$ and performing gradient queries to update $G$. After the training is done, generator $G$ is used to supply small-loss instances $(\boldsymbol{x}_f, c)$ to train the surrogate model (the unknown part). We observe a serious mode collapse problem during the GAN training. So we utilize the distance-aware training introduced in Yang et al. [2019], to encourage the c-GAN to generate more diverse small-loss instances. In the new method, the training of the generator is not based on a min-max game, or on traditional GAN training. Instead, it simply trains the generator toward minimizing cross-entropy loss. While the generator $G$ fails to generate natural-looking inputs even upon convergence, it generates abundant small-loss instances for every label, and divergence loss helps it generate a good variety of outputs. During SFL training, the generator can adjust itself to the changing server-side model.

**Gradient matching model extraction (GM-ME).** Gradient matching (GM) in ME attack has been investigated in Jagielski et al. [2020], Milli et al. [2019] and is used in combination with prediction query to improve the extraction performance. Since, in SFL, prediction query is not allowed, we navigate this strict threat model's restriction by adopting gradient matching (GM) loss. For a given label $\boldsymbol{y}_i$, GM loss has the following form:

$$\mathcal{L}_{GM} = |\nabla_{\boldsymbol{x}_i}\mathcal{L}(S(C(\boldsymbol{x}_i)), \boldsymbol{y}_i) - \nabla_{\boldsymbol{x}_i}\mathcal{L}(V(C(\boldsymbol{x}_i)), \boldsymbol{y}_i)|_2^2 \tag{1}$$

where, $\boldsymbol{x}_i$ denote inputs, $C$ denotes client-side model, $S$ and $V$ denotes the surrogate model and victim model, respectively. For each input, attacker would query gradients with different label $\boldsymbol{y}_i$ to get as much information as possible. This attack performs extremely well for small $N$ but degrades significantly for a larger $N$. Its performance also depends on the domain similarity between the auxiliary dataset and the victim dataset.

## 4.2 ME attacks with training data.

Next we discuss the case when the attacker has a subset of training data, corresponding to the strongest data assumption .

**Training-based model extraction (Train-ME).** For attackers with a subset of the training data, derivation of an accurate surrogate model can be done using supervised learning (through minimizing the cross entropy loss on the available data). We call this Train-ME, similar idea is also adopted in Fu et al. [2022] to extract the entire model of the other party. Train-ME only relies on the white-box assumption of the client-side model, using it to initialize the surrogate model and does not need to use the gradient query at all. Surprisingly, it is one of the most effective ME attacks.

**Gradient-based soft label training model extraction (Soft-train-ME).** If gradient query is allowed and a subset of training data is available, the attacker can achieve better ME attack performance compared to Train-ME. To utilize gradients, a naive idea is to combine the GM loss with cross-entropy loss in Train-ME. However, our initial investigation shows they are not compatible; the cross-entropy loss term usually dominates and the GM loss even hurts the performance. An alternative approach is to use soft label. We build upon the method in Gu et al. [2020] which shows that gradient information of incorrect labels is beneficial in knowledge distillation, and use it for surrogate model training. Specifically, for each input $\boldsymbol{x}_i$, gradients of the ground truth label as well as incorrect labels are collected (a total $N_C$, where $N_C$ is the number of classes). For an input $\boldsymbol{x}_i$ with true label $c$, its soft label $q_i^k$ of $k$-th ($k \neq c$) label is computed as follows:

$$q_i^k = (1 - \alpha) * \frac{cos(\boldsymbol{e}^k, \boldsymbol{e}^c)}{\sum_{m=1, m \neq c}^{N_C}(cos(\boldsymbol{e}^m, \boldsymbol{e}^c) + 1)} \tag{2}$$

where, $\boldsymbol{e}^k$ denotes flattened gradients of label $k$, $q_i^k$ denotes soft label for the k-th label $k$ and $\alpha$ is a constant ($\alpha > 0.5$). The derived $(\boldsymbol{x}_i, \boldsymbol{q}_i)$ pair is then used in the surrogate model training in addition to the true label $c$ (which is the only difference from the Train-ME).

# 5 Model Extraction Performance

In this section, we demonstrate the performance of the proposed ME attacks and the baseline attack on SFL schemes. All experiments are conducted on a single RTX-3090 GPU. For the SFL model training, we set the total number of epochs to 200, and use SGD optimizer with a learning rate of 0.05 and learning rate decay (multiply by factor of 0.2 at epochs 60, 120 and 160). We assume all clients participate in every epoch with an equal number of training steps. To perform ME attacks, the attacker uses an SGD optimizer with a learning rate of 0.02 to train the surrogate model and we report the best accuracy and fidelity. We evaluate accuracy of the surrogate model on the validation dataset. We use the label agreement as fidelity, defined as the percentage of samples that the surrogate and victim models agree with over the entire validation dataset, as in Jagielski et al. [2020]. Detailed settings for each ME attack are described in Appendix A.1.

## 5.1 ME attack on SFL with fine-tuning based training

We first perform the proposed ME attacks on fine-tuning SFL version with consistent gradient query. Here we use a pre-trained model and set the number of gradient queries to 100K. On a victim VGG-11 model on CIFAR-10 dataset, whose original accuracy is 91.89%, performance of all five ME attacks are shown in Table 2. For each of the ME attacks, we vary hyper-parameters and report the one that achieves the best attack performance; details on different hyper-parameters are included in Appendix A.3. When $N = 2$, all five ME attacks can achieve near-optimal accuracy and fidelity performance. For Craft, GAN and GM ME, the accuracy drops to around 80% when $N$ is 5, it sharply drops to below 40% when $N$ is 8. For Train and SoftTrain ME, accuracy slightly degrades when $N$ is 5, and reduces to around 70% when $N$ is 8. These results show that ME attack performance strongly correlates with the #layers in server-side model. With an increasing #layers in server-side model, ME attack performance reduces as the extraction problem becomes harder with more unknown parameters and more complicated input feature space.

**Observation.** Different attacks present different and interesting characteristics. Craft-ME has a steady attack performance and can succeed even with a tight query budget. GAN-ME needs a large query budget to train the c-GAN generator towards convergence but can achieve better accuracy and fidelity than Craft-ME for $N \leq 5$. GM-ME requires an auxiliary dataset that is similar to the training data. If CIFAR-100 is used to attack CIFAR-10 model, GM-ME achieves almost perfect extraction for small $N$. However, its performance degrades if MNIST or SVHN are used as auxiliary datasets. For a high $N$, the surrogate model fails to converge on the GM loss, and its extraction performance suffers from a sharp drop. For attacks with training data such as Train and SoftTrain MEs, both accuracy and fidelity are much higher than attacks without training data. When $N \geq 6$, SoftTrain-ME can achieve slightly better accuracy and fidelity than Train-ME.

Table 2: ME attack performance on SFL on fine-tuning and training-from-scratch applications. The victim is a VGG-11 model on CIFAR-10 with 91.89% validation accuracy. For Train, SoftTrain and Naive baseline, for the fine-tuning setting, data assumption is 1K training data (randomly sampled), and for the train-from-scratch setting, the number of clients is 10 and each client has 5K training data.

| Metric | $N$ | Fine-tuning | | | | | | Training-from-scratch | | | | | |
|---|---|---|---|---|---|---|---|---|---|---|---|---|---|
| | | Craft | GAN | GM | Train | SoftTrain | Naive | Craft | GAN | GM | Train | SoftTrain | Naive |
| Accuracy (%) | 2 | 91.64 | 91.86 | 92.02 | 92.05 | 91.99 | | 85.99 | 85.99 | 53.06 | 90.58 | 90.31 | |
| | 5 | 83.46 | 84.93 | 80.28 | 90.82 | 90.48 | 49.64 | 35.58 | 40.03 | 12.13 | 89.86 | 87.02 | 72.63 |
| | 8 | 35.48 | 18.82 | 12.45 | 70.28 | 71.32 | | 15.34 | 17.49 | 10.88 | 78.64 | 56.78 | |
| Fidelity (%) | 2 | 98.23 | 98.42 | 99.87 | 99.29 | 99.10 | | 92.37 | 89.59 | 54.63 | 99.34 | 98.87 | |
| | 5 | 86.32 | 87.49 | 84.33 | 94.84 | 94.67 | 50.62 | 41.32 | 38.72 | 11.87 | 95.40 | 89.83 | 72.62 |
| | 8 | 36.11 | 18.62 | 12.63 | 71.79 | 72.45 | | 15.63 | 17.44 | 10.67 | 80.01 | 57.78 | |

## 5.2 ME attack on training-from-scratch SFL

Next, we investigate the proposed ME attack performance in training-from-scratch SFL case. A good attack-time-window for gradient-based ME attacks is at the end of training when gradients do not vary as much and the model converges. So for Craft, GAN and GM-ME, we launch the attack at epoch 160 to get more consistent gradients. As the model is updated by multiple clients, the percentage of

malicious clients also affects the ME attack performance. We found that with more benign clients (or fewer malicious inputs), the server-side model returns more consistent gradients to the attacker. Attack performance for three attacks are shown in Table 2 for 10-client SFL training-from-scratch case. Results with other hyperparameter settings are included in Appendix A.4.

**Observation.** Because of the poisoning effect of malicious inputs sent by the attacker in gradient-based attacks, in all the methods except Train-ME, the victim model's accuracy suffers from a 2-3% degradation, resulting in a less accurate surrogate model. For gradient-based attacks without training data (Craft, GAN and GM MEs), we notice a sharp ($> 3\%$) performance drop in both accuracy and fidelity compared to the consistent query case. The sharp drop in ME attack performance is caused by *inconsistent gradients*. Take Craft-ME as an example. Crafted inputs that have a small loss at an earlier epoch of the training can have a large loss in the final model. Training surrogate model with large amount of inconsistent information results in poor training accuracy. Compared to Craft-ME, GAN-ME is more robust to inconsistent gradients as the generator can adjust itself to the change of server-side model, resulting in a better extraction performance. However, when $N$ is larger, the generator does not converge well and its performance drops drastically. Last but not the least, GM-ME completely fails with inconsistent gradients, even for small $N$. This implies that the GM loss is super sensitive to inconsistent gradients and is only effective in consistent query cases. A comparison of SoftTrain-ME and Train-ME shows that SoftTrain's advantage diminishes because of the poisoning effect. Thus, Train-ME is more effective for an attacker with training data in training-from-scratch SFL.

### 5.3   ME attack performance for other architectures and datasets

We perform extensive analysis on the attack performance of the proposed ME schemes on other architectures and datasets. We use Train-ME attack, the best performer for both fine-tuning and training-from-scratch cases and use the accuracy as the performance measure. To test ME attack on different datasets, we perform Train-ME attack with 1K training data on VGG-11 model with $N$ set to 5, on MNIST [LeCun, 1998], FMNSIT [Xiao et al., 2017], SVHN [Netzer et al., 2011] and CIFAR-100 datasets [Krizhevsky et al., 2009] (in addition to CIFAR-10). As shown in Fig. 4 (a), for all datasets except CIFAR-100, ME attack achieves accuracy very close to the original. For CIFAR-100, the extracted accuracy is over 20% below the original because of its task difficulty. Additionally, we also test Train-ME performance with 2% and 20% ImageNet training data of Mobilenet-V2 on ImageNet dataset. As shown in Fig. 4 (b), ME attack fails badly due to the complexity of ImageNet dataset [Deng et al., 2009], resulting in a high accuracy gap of 10% when $N$ is set to 2. For different architectures, we choose Resnet-20, Resnet-32 [He et al., 2016] and Mobilenet-V2 [Sandler et al., 2018] on CIFAR-10 dataset (with necessary adaptations). For Resnet and Mobilenet family, we assign last 4 layer-blocks and 1 FC layer to server-side model. As shown in Fig. 4 (c), with the same proportion of layers (5 out of 11) being assigned to server-side model, ME attack is much less effective on Resnet-20 than on VGG-11. A comparison of the performance of Resnet-32 and Mobilenetv2 with similar proportion of layers being assigned to server-side (5 out of 17 and 20, respectively), ME on Resnet-32 is also much worse than on MobilenetV2. This indicates Resnet architecture is more resistant to ME attack.

## 6   Discussion

Our evaluation showed that ME attack performance drops with increasing #layers in server-side model. Thus, a simple idea to improve resistance to ME attack is to use a larger $N$. However this implies that the #layers in client-side model would be smaller, thereby undermining clients' data privacy. Data privacy in SFL can be represented by the Mean Square Error (MSE) performance of an MI attack as outlined in Li et al. [2022]. The implementation detail is included in Appendix B.1. As shown in Fig. 4 (d), extracted accuracy decreases with a larger $N$, and MSE decreases too. This means MI attack can provide more precise reconstruction which compromises data privacy. We will extensively consider the tradeoff between ME resistance and data privacy in future.

**Defense Method.** We also investigate defensive methods against ME attacks on SFL. According to the analysis in Fig. 2, extraction is easier because client-side model is leaked to the attacker. Thus, to make ME attacks more difficult, we can restrict the client-side model's feature extraction capabilities. Towards this goal, we apply L1 regularization with three different strength ($\lambda$ = 5e-5, 1e-4 and 2e-4)

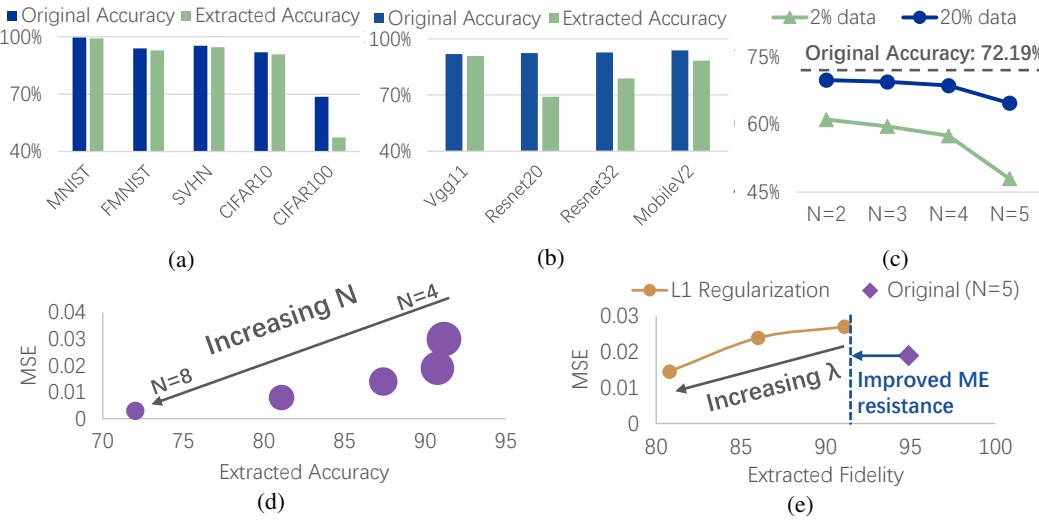

Figure 4: **Top row:** Extensive results on model extraction (ME) attacks. (a) ME attack performance of VGG-11 on other datasets. (b) ME attack performance of MobilenetV2 on ImageNet. (c) ME attack performance of other architectures on CIFAR-10 dataset. **Bottom row:** (d) Tradeoff between ME resistance and data privacy (MSE). (e) L1 regularization as effective defense for ME attacks.

on the client-side model to penalize its weight magnitude. As shown in Fig. 4 (e), this simple defense effectively improves the resistance to ME attack (specifically, Train-ME with 1K data) without hurting data privacy. While there is some accuracy degradation, as shown in Appendix A.5, this demonstrates the potential of using regularization to defend ME attacks.

## 7 Ablation Study

**ME attack with non-IID data.** We consider the non-IID (independent and identically distributed) case where the attacker only has data from $C$ classes of CIFAR-10. Results presented in Appendix A.6 show that ME attack performance is still good for $C = 5$ but degrades sharply when $C = 2$.

**Adversarial attack based on successful ME attack.** As mentioned before, the goal of ME attack is to launch more successful adversarial attacks. We perform transfer adversarial attacks using a surrogate model extracted by the strongest Train-ME attack. As shown in Appendix A.7, SFL with proper $N$ achieves better resistance to adversarial attacks.

**ME attack without architecture information.** In Appendix A.8, we investigate simple variants (longer, shorter, wider, and thinner) of the original architecture as the surrogate model architecture. We find that the performance of ME attacks is similar for the different architectures – the exception is GM-ME which fails for different surrogate architectures.

## 8 Conclusion

In this work, we study the model IP protection capability of SFL and its resistance to model extraction (ME) attack. We propose five viable ME attack methods for the threat model where gradient query is allowed but prediction query is not allowed. For the case when there are enough number of layers in server-side model, model IP can be protected well and transfer adversarial attack is not successful. However, data privacy could be compromised and so this factor needs to be considered in the development of such schemes. We also point out a possible way of defending such attacks through regularization and plan to expand on it in the near future.

**Broader Impact.** This work points out the vulnerability of Split Federated Learning (SFL), to model extraction attacks, and should prevent a naive adoption of SFL as a model IP protection method. We believe that the attacks presented here would initiate research in the development of defense schemes to mitigate such attacks, help design a more robust SFL and possibly help in the design of neural network models that are inherently resilient to such attacks.

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
