# OpenReview forum: "Model Extraction Attacks on Split Federated Learning"
_NeurIPS.cc/2022/Conference — NeurIPS 2022 Submitted_

### Official Review · Reviewer_sNJB · 2022-07-10

**Rating:** 4
**Confidence:** 5
**Soundness:** 2 fair
**Presentation:** 3 good
**Contribution:** 3 good

**Summary:**

This paper investigates model extraction attacks on split federated learning. The paper assumes that the attacker cannot be able to access the model predictions from the server. Hence, the paper leverages the gradients to extract the server-side model. Five model extraction attack methods are proposed for different data assumptions.

**Questions:**

Please clarify the no prediction assumption (weakness 1) and the no data assumption (Weakness 3).

**Limitations:**

The authors addressed the limitations and potential negative societal impact.

**Strengths And Weaknesses:**

Strengths:
1. The paper investigates five model extraction attacks against SFL.

2. The evaluation considers several interesting scenarios, such as non-iid data distributions, adversarial attacks, and attacks without knowing model architectures.

Weaknesses:
1. The paper claims that the client-side attacker cannot use the existing model extraction attacks in the split federated learning because the client cannot get the predictions from the server. However, in the inference phase, the clients can get the predictions. The proposed attacks can be further compared with the existing model extraction attacks through predictions in the inference phase.

2. The novelty of this paper is limited. Most proposed attacks leverage the existing model extraction attacks. Train-ME is the most effective attack. However, Train-ME is a very straightforward attack, which does not even need the gradient query. This may indicate that the model extraction attacks in SFL is not a trivial problem.

3. The no data assumption is confusing. If the attacker is a client, the attacker should at least have access to the client’s data.

4. It would great if the proposed attacks could be compared with some baselines. For example, since the attacker is one of the clients, what is the attack performance when the attacker uses the client’s data to train a model?

5. The experiment only considers 10 clients. However, in the real-world settings, the number of clients should be much large. What’s the performance of the proposed attacks when a large number of clients participate in the SFL?

6. The proposed attacks do not perform well on complex datasets (e.g., ImageNet, CIFAR100) and complex models (e.g., ResNet).

7. The scope of the paper is pretty narrow. The proposed attacks can be only applied to split federated learning.

---

> ### Author Response · Authors · 2022-08-02
> **Response to Reviewer sNJB (1/2)**
>
> **Q1**: The paper claims that the client-side attacker cannot use the existing model extraction attacks in the split federated learning because the client cannot get the predictions from the server. However, in the inference phase, the clients can get the predictions. The proposed attacks can be further compared with the existing model extraction attacks through predictions in the inference phase.
>
> **A1**: Thanks for pointing out this concern on our threat model. Below we list several reasons to justify our assumptions that clients cannot access the prediction:
> The SFL scheme in [Thapa, 2022], on which this work is based, does not provide clients prediction results and so we too assume the same.
> There exist many systems that do not expose prediction API to customers. For example, Google keyboard uses an FL-trained model as part of the next-word prediction service, where the final suggested word is a combination of the FL-trained model with another model that is hidden from the customer (refer to [1] by Google). Thus users do not know the exact prediction of the FL model, that is built internally. Similarly, we believe that the prediction result of the SFL-learned classification engine should be wrapped up internally to improve companies’ service quality, instead of exposing directly to customers.
> Should it be possible for prediction to be accessed, the model extraction (ME) problem in SFLwould become an easier problem (sub-problem)  compared to ME on prediction API, which has been thoroughly investigated in [2, 3]. As we demonstrate in Fig.3, existing methods can succeed in ME with high accuracy and fidelity. For the server (SFL scheme and model owner), it makes sense not to expose the prediction.
>
> References
> [1] T Yang et al. - 2018 - Applied Federated Learning: Improving Google Keyboard Query Suggestions
> [2] F Tramer et al. - 2016 - Stealing Machine Learning Models via Prediction APIs
> [3] M Jagielski et al. - 2020 - High accuracy and high fidelity extraction of neural networks
>
>
>
> **Q2**: The novelty of this paper is limited. Most proposed attacks leverage the existing model extraction attacks. Train-ME is the most effective attack. However, Train-ME is a very straightforward attack, which does not even need the gradient query. This may indicate that the model extraction attacks in SFL is not a trivial problem.
>
> **A2**:  The major contribution of this paper is exposing the vulnerability of SFL. We propose five attacks that can get a very accurate and high-fidelity surrogate model. Thus the claim that baseline SFL provides IP protection is no longer true! On a more positive note, we also show how resistance can be improved by leaving more layers on the server-side, and by applying regularization techniques during training.
>
> **Q3**: The no data assumption is confusing. If the attacker is a client, the attacker should at least have access to the client’s data
>
> **A3**:
> Our “no data” assumption means the attacker has no data at all or randomly generated noise data. We have elaborated on this in the introduction. For the case when the attacker has training data, we have discussed corresponding attacks in section 4.2.
>
> **Q4**: It would great if the proposed attacks could be compared with some baselines. For example, since the attacker is one of the clients, what is the attack performance when the attacker uses the client’s data to train a model?
>
> **A4**: Thanks for the suggestion.  We performed the naive baseline attack (uses client’s data to train a model) for VGG-11 architecture on CIFAR-10, assuming that the attacker has only 2% training data (1K data).  The table below shows that the accuracy/fidelity performance is much lower compared to model extraction attacks.
>
> | 1K data  | Naive Baseline | Train-ME (N=2) | SoftTrain-ME (N=2) | Train-ME (N=8) | SoftTrain-ME (N=8) |
> |--------------|:---------------:|:---------------:|:------------------:|-----------------|--------------------|
> | Accuracy |      49.64     |      92.05     |        91.99       | 70.28          | 71.32              |
> | Fidelity    |      50.62     |      99.29     |        99.10       | 71.79          | 72.45              |
>
> We have included the simple baseline results in Table 2 for both finetuning and training cases. We have also listed differences between the naive baseline and proposed attacks in Table 1.
>
> **Q5**: The experiment only considers 10 clients. However, in the real-world settings, the number of clients should be much large. What’s the performance of the proposed attacks when a large number of clients participate in the SFL?
>
> **A5**:  Please see our response in part2.

---

> ### Author Response · Authors · 2022-08-02
> **Response to Reviewer sNJB (2/2)**
>
> **Q5**: The experiment only considers 10 clients. However, in the real-world settings, the number of clients should be much large. What’s the performance of the proposed attacks when a large number of clients participate in the SFL?
>
> **A5**:  Sorry for causing confusion. The number of client settings is actually different for fine-tuning and training-from-scratch cases. We have revised the description to make it more clear.
>
> For the fine-tuning case, we simulate 50 clients by using 1K data per client. For the training-from-scratch case, we have results for 5-client and 10-client in Appendix A.4. Here we see that increasing the #clients generally reduces the model extraction performance since the number of queries per data is less. Unfortunately, we do not have simulation results for more than 10 clients.  We will certainly investigate the scalability problem in our future work.

---

> > ### Comment · Reviewer_sNJB · 2022-08-08
> > **Q2 and Q3**
> >
> > I appreciate the author's responses and revisions. Although most of my questions have been addressed, I still have two concerns.
> >
> > First, the novelty of the paper is limited (Q2). Federated learning has been shown vulnerable in many recent works. Showing the vulnerability of split federated learning is not a major finding.
> >
> > Second, the data assumption (no data or auxiliary data) is not practical (Q3). Since the attacker is assumed as a participant client, the attacker should have access to the in-distribution data.

---

> > > ### Author Response · Authors · 2022-08-09
> > > **Response to "Q2 and Q3" comments from Reviewer sNJB**
> > >
> > > **Response to Q2**: We kindly disagree that showing the vulnerability of split federated learning is not a major finding. The model extraction attack (MEA) we are focusing on is a unique attack vector for SFL that does not exist for  FL. In FL, all participating clients can get access to the entire model, thus performing MEA is meaningless as clients already have the full model. But in SFL, clients cannot get access to the entire model, but only the client-side model, which is a small part of the model (before the cut-layer). This is why SFL was typically seen as an “IP-protected” FL. In this paper, we point out the vulnerabilities of SFL and refute the claim of SFL being secure from MEA.
> > >
> > > **Response to Q3**: In this paper, we consider all three data assumptions, namely, no-data, out-of-distribution data and in-distribution data; Section 4 describes the attacks in each case. The no-data case is justified since **participants do not have to have training data**. Participants can be free-riders [1-2], that is, they claim to have abundant data but actually do not. Such a case exists since FL/SFL does not have an effective checking mechanism and everyone can participate. Our “no data” setting corresponds to the free-rider setting and can include cases where the client has no data at all or just randomly generated noise data.
> > >
> > > Reference
> > >
> > > [1] Y Fraboni et al., Free-rider Attacks on Model Aggregation in Federated Learning, PMLR 2021
> > >
> > > [2] L Lyu et al., Collaborative fairness in federated learning, Federated Learning, Springer 2020

---

> ### Author Response · Authors · 2022-08-08
> **A Letter to Reviewer sNJB**
>
> Dear reviewer,
>
> Thank you for the detailed review.
>
> We hope that our responses addressed all your concerns. We would be very happy to address
> followup questions if any.
>
> Looking forward to hearing back from you.
>
> Best,
> Authors

---

### Official Review · Reviewer_HjPo · 2022-07-11

**Rating:** 6
**Confidence:** 4
**Ethics Flag:** Yes
**Soundness:** 3 good
**Presentation:** 3 good
**Contribution:** 3 good

**Summary:**

Federated learning is a distributed learning paradigm that allows multiple clients and a server to collaboratively learn a machine learning model without data sharing. However, federated learning still suffers from model privacy concerns, since the server and the clients all have access to the full global model. Split federated learning is proposed as a solution to both communication reduction and privacy enhancement. In split federated learning, the machine learning model is split into two parts, one held by the server and the other one held by the clients. Since the clients have access to only part of the model, split federated learning was considered to be robust to client-side model extraction attacks. However, in this work, the authors find that this is a false sense of security. Specifically, a malicious client can perform model extraction attacks to learn a surrogate model that have similar accuracy/predictions compared to the target server-side model.

**Questions:**

- What is the accuracy of the model obtained via the aforementioned baseline, i.e., locally trained by the attacker with the available training data? Do the proposed attacks outperform it?
- What is the complexity of the proposed attacks?
- Can existing statistics-based Byzantine-robust FL methods defend against the attacks?
- Is there any possible reason why ResNet shows better robustness to ME attack?

**Ethics Review Area:**

["Privacy and Security (e.g., consent)"]

**Limitations:**

The negative societal impact may need to be stated more explicitly, e.g., via adding a paragraph in the discussion.

**Strengths And Weaknesses:**

### Strengths
- Five different model extraction attacks are proposed in this work. The authors consider attack different settings for the attack, e.g., whether the attacker (malicious client) has local training data.
- The authors conduct extensive experiments to show the effectiveness of their model extraction attacks. It is impressive that the attacks do work in many cases and it is interesting to see the impact of different parameters, e.g., the number of queries, the number of layers in the server-side model, and model architectures.
- The paper is generally well-written and easy to follow.

### Weakness
- When evaluating the model extraction attacks with training data available, it is intuitive to compare the attacks with a simple baseline that uses the training data to train a model locally on the malicious client. I would expect the authors to perform a simple comparison with such baseline. This is essential because if the attacker can locally train a better model, then there is no motivation for the attacker to perform the model extraction attacks.
- There is no analysis on the complexity of the proposed attacks. It is interesting to see some analysis on the attack cost, e.g., $O()$ notations with respect to the number of parameters in the server-side model.
- There are some defenses in federated learning that leverages statistical analysis of the local gradients (known as Byzantine-robust FL methods). I am curious whether these methods are sufficient to defend against the proposed attacks.
- In Section 5.3, it is claimed that "This indicates Resnet architecture is more resistant to ME attack". It would be interesting to see some explanation on why this is the case.
- The potential societal impact may need to be stated more explicitly.
- Writing issues. On line 3 of Algorithm 1, I would say it is better to average model parameters (i.e., $W$) instead of models (i.e., $C$). In Figure 4, the captions of subfigures mismatch.

---

> ### Author Response · Authors · 2022-08-02
> **Response to Reviewer HjPo (1/2)**
>
> We merged description under weaknesses and questions for better readability:
>
> **Q1**: When evaluating the model extraction attacks with training data available, it is intuitive to compare the attacks with a simple baseline that uses the training data to train a model locally on the malicious client. I would expect the authors to perform a simple comparison with such baseline. This is essential because if the attacker can locally train a better model, then there is no motivation for the attacker to perform the model extraction attacks. What is the accuracy of the model obtained via the aforementioned baseline, i.e., locally trained by the attacker with the available training data? Do the proposed attacks outperform it?
>
> **A1**: Thank you for the suggestion. We performed the naive baseline attack for VGG-11 architecture on CIFAR-10 and CIFAR-100, assuming that the attacker has only 2% training data (1K data).  The table below shows that the accuracy/fidelity performance is much lower compared to model extraction attacks.
>
> | 1K data  | Naive Baseline | Train-ME (N=2) | SoftTrain-ME (N=2) | Train-ME (N=8) | SoftTrain-ME (N=8) |
> |--------------|:---------------:|:---------------:|:------------------:|-----------------|--------------------|
> | Accuracy |      49.64     |      92.05     |        91.99       | 70.28          | 71.32              |
> | Fidelity    |      50.62     |      99.29     |        99.10       | 71.79          | 72.45              |
>
> We included the simple baseline results in Table 2 for both finetuning and training cases. We also listed differences between the naive baseline and proposed attacks in Table 1.
>
> **Q2**: There is no analysis on the complexity of the proposed attacks. It is interesting to see some analysis on the attack cost, e.g., O() notations with respect to the number of parameters in the server-side model.
>
> **A2**: Thank you for raising this concern.
> We did not include the query complexity analysis because our proposed methods are not based on cryptanalytic model extraction (such as Carlini et al. - Cryptanalytic Extraction of Neural Network Models). Our attacks are based on approximate model extraction where we can better approximate the target model with more queries, but performance may vary from case to case.
>
> So here we include time measurement results to provide a sense of how much time it takes to deploy proposed attacks. We show the time for preparation and surrogate model training of all five attacks. For example, preparation phase includes crafting inputs in Craft-ME or crafting soft labels in SoftTrain-ME. The measurement is done using Python time library, on a desktop PC equipped with a R7-5800X CPU and a single RTX-3090 GPU.
>
> | Time Cost (s)         | Craft   |  GAN |   GM    | Train | SoftTrain |
> |--------------------------|:--------:|:-------:|:----------:|:-------:|:------------:|
> | Preparation            | 317.8 |  44.5  |  30.7    |  4.7    |    18.9     |
> | Surrogate Training | 381.2 | 339.3 | 5523.1 | 313.4 |   949.5    |
> | Total                       | 699.0 | 383.8 | 5553.8 | 318.1 |   968.4    |
>
> More details have been included in a new section in Appendix A.9 in the revised paper.
>
> **Q3**: There are some defenses in federated learning that leverages statistical analysis of the local gradients (known as Byzantine-robust FL methods). Can existing statistics-based Byzantine-robust FL methods defend against the attacks?
>
> **A3**: Four of our proposed attacks require poisoning the model - “sending false input” or “correct inputs with false labels”, and the Byzantine-robust FL method is known to be robust to the poisoning effect. However, we are not sure whether these methods will be applicable to SFL and will be able to circumvent gradient-based ME. Thank you for providing this lead. We will investigate this in the near future.
>
> **Q4**: In Section 5.3, it is claimed that "This indicates Resnet architecture is more resistant to ME attack". It would be interesting to see some explanation on why this is the case. Is there any possible reason why ResNet shows better robustness to ME attack?
>
> **A4**: Our intuitive understanding of why ResNet is more resistant compared to VGG architecture is because of its residual connection which allows raw input to be passed to the later layers. In VGG architecture, front layers tend to play an important role as they decide what features will feed to later layers. But in ResNet, later layers can receive both the extracted feature and the raw inputs, thus, to some degree, reducing the importance of front layers that are public to the attacker.
> We would like to perform a deeper analysis in our future work with the hope that we can derive special architectures which have a natural resistance against the proposed attacks.
>
> **Q5**: The potential societal impact may need to be stated more explicitly.
>
> **A5**: Please see part2 of our response.

---

> > ### Comment · Reviewer_HjPo · 2022-08-10
> > **Thanks for the response**
> >
> > I appreciate the authors' detailed response. I will keep my rating for this work.

---

> ### Author Response · Authors · 2022-08-02
> **Response to Reviewer HjPo (2/2)**
>
> **Q5**: The potential societal impact may need to be stated more explicitly.
>
> **A5**: Thanks for the suggestion. We strongly agree that pointing out the societal impact is very important for attack work. We add a “Broader Impact” subsection which states the societal impact of this work: “This work points out the vulnerability of Split Federated Learning (SFL), to model extraction attacks, and should prevent a naive adoption of SFL as a model IP protection method. We believe that the attacks presented here would initiate research in the development of defense schemes to mitigate such attacks,  help design a more robust SFL and possibly help in the design of neural network models that are inherently resilient to such attacks.”

---

### Official Review · Reviewer_WXHo · 2022-07-11

**Rating:** 6
**Confidence:** 3
**Soundness:** 3 good
**Presentation:** 3 good
**Contribution:** 3 good

**Summary:**

The authors perform five different model extraction attacks in settings of split federated learning (SFL). Such attacks exploit gradient information from the server-side to conduct model extraction attacks. Besides, this paper proposed an approach to obtain the necessary shadow data for constructing attackers.

**Questions:**

  1. Craft-ME method, both the surrogate model and input x_r are randomly initialized, and the surrogate model generates x_r since the server-side model won't provide loss respect with x_r. Even though we can generate plausibly (x_r, c) labels via such a random surrogate model, how can these collected (x_r, c) instances be applied to train the surrogate model well?
  2. Gradient matching model extraction (GM-ME): since the prediction query is disallowed, V is unknown in this setting. how to compute \nabla_{x_i} L(V(C(x_i)), y_i) (see line 196, eq. 1)? Normally, Victim part will only compute gradient with respect to C's party weights.
  3. Training-based model extraction (Train-ME): how much training data are used to train such a surrogate model, compared with the data size in SFL settings.

**Limitations:**

  1. Gradient matching model extraction (GM-ME) relies on the gradient of x_i, which is infeasible in reality.
  2. Training-based model extraction (Train-ME)'s idea is done by previous work, using model transferability. see [1]

[1] Fu, Chong, et al. "Label inference attacks against vertical federated learning." 31st USENIX Security Symposium (USENIX Security 22), Boston, MA. 2022.

**Strengths And Weaknesses:**

** strengths
  1. Give a comprehensive investigation of model extraction attacks in split federated learning settings, and provide rich experimental results.
** weaknesses
  1. Different attack approaches are explained in an abstract way. It is hard to understand the attack assumption, how to craft data, how the attacker works, etc.

---

> ### Author Response · Authors · 2022-08-02
> **Response to Reviewer WXHo**
>
> **Q1**: Different attack approaches are explained in an abstract way. It is hard to understand the attack assumption, how to craft data, how the attacker works, etc.
>
> **A1**:  We apologize that the attack model was not clear! The attack assumptions are discussed in detail in section 3.1 and methodologies for each of the proposed attacks are separately introduced in section 4. To better explain the procedure of all five attacks, we have added a diagram in Appendix A.2 to help show the process of extraction attacks. We will release our code publicly as well.
>
> **Q2**: Craft-ME method, both the surrogate model and input x_r are randomly initialized, and the surrogate model generates x_r since the server-side model won't provide loss respect with x_r. Even though we can generate plausibly (x_r, c) labels via such a random surrogate model, how can these collected (x_r, c) instances be applied to train the surrogate model well?
>
> **A2**: In our assumption, clients can get access to $\nabla_{A} L$, and have access to the client-side model C. Given $A=C(x_r)$, clients can apply the chain rule ($\nabla_{x_r} L = \frac{\partial L}{\partial A} \frac{\partial A}{\partial x_r} = \nabla_{A} L \frac{\partial C(x_r)}{\partial x_r}$) to acquire $\nabla_{x_r} L$, which can be used to update $x_r$ to make the loss small. Once some pairs of ($x_r$, c) are generated, these small-loss instances are used as normal data to train the surrogate model using cross-entropy loss.
>
> **Q3**: Gradient matching model extraction (GM-ME): since the prediction query is disallowed, V is unknown in this setting. how to compute \nabla_{x_i} L(V(C(x_i)), y_i) (see line 196, eq. 1)? Normally, Victim part will only compute gradient with respect to C's party weights.
>
> **A3**: Actually, even if V is not known,  \nabla_{x_i} L can still be accessed. Once the server calculates L, it performs backward propagation on V and sends the $\nabla_{C(x_i)} L$ back to the client (the detailed process is illustrated in Algorithm 1, line 12). The client can then apply the chain rule (described in A2) using client model C and $\nabla_{C(x_i)} L$ to acquire gradients on $x_i$.
>
> **Q4**: Training-based model extraction (Train-ME): how much training data are used to train such a surrogate model, compared with the data size in SFL settings.
>
> **A4**: Sorry for not clearly describing the data setting. We have added the necessary descriptions in the caption of Table 2. We use 1K training data (randomly sampled) for the fine-tuning cases. For training-from-scratch experiments, we use a 10-client SFL scheme where each client (including the malicious one) has 5K training data.
>
> **Q5**: Gradient matching model extraction (GM-ME) relies on the gradient of x_i, which is infeasible in reality.
>
> **A5**: Please see the responses under **A2** and **A3**.
>
> **Q6**: Training-based model extraction (Train-ME)'s idea is done by previous work, using model transferability. see [1]: [1] Fu, Chong, et al. "Label inference attacks against vertical federated learning." 31st USENIX Security Symposium (USENIX Security 22), Boston, MA. 2022.
>
> **A6**: Thanks for pointing this out. At the time of submitting the paper in May, we were not aware of this USENIX conference paper. We have included it in the updated version and pointed out the similarity between our Train-ME and the “model completion” technique in [1].

---

### Official Review · Reviewer_HaF2 · 2022-07-11

**Rating:** 6
**Confidence:** 3
**Soundness:** 3 good
**Presentation:** 3 good
**Contribution:** 3 good

**Summary:**

The authors expose the vulnerability of split Federated Learning and show how model extraction attacks can be launched by malicious clients querying the gradient information from server-side.

They proposed five different variants of model extraction attacks (ME) on split federated learning. The attacks use different gradient schemes, including data crafting, data generating, gradient matching and soft label crafting. They also made different assumptions for the data such as no data, only auxiliary data (out-of-distribution data) and training data (in-distribution data). They interestingly find that in a 5-layer-in-server SFL, an ME attack can derive a surrogate model with over 90% accuracy, and less than 2% accuracy degradation.

For the experiment they tried both fine-tuning the SFL models and training from scratch for different gradient consistency.
They showed that the ME attacks can succeed even without any data when fine-tuning. They also concluded that the ME attacks would succeed better with the increase of layers on the server side.

**Questions:**

What do authors think about their possible way of defending such attacks through regularization? How good is that?

Have the authors also looked the LEAF datasets for the experiments? LEAF seems to be the standard dataset for FL papers, with natural client partitions and other FL constraints.

**Limitations:**

The defense mechanism is quite basic and it would be nice to see some strong defense mechanisms with more analyses and numerical experiments

**Strengths And Weaknesses:**

Strengths:
- It’s an original work and perhaps the first one on the possible attacks on SFL systems.
- Quality of the work is good, and the writing it’s quite clear.

Weaknesses:
- Some related work is missing. The authors mention that Split Learning is firstly introduced in reference "[Gupta and Raskar, 2018]" while the idea goes further back and there are already several papers on the field. For example (non exhaustive list):
* Kang, Yiping, et al. "Neurosurgeon: Collaborative intelligence between the cloud and mobile edge." ACM SIGARCH Computer Architecture News 45.1 (2017): 615-629.
* Yousefpour, Ashkan, et al. "Guardians of the deep fog: Failure-resilient DNN inference from edge to cloud." Proceedings of the First International Workshop on Challenges in Artificial Intelligence and Machine Learning for Internet of Things. 2019.
* Liu, Peng, Bozhao Qi, and Suman Banerjee. "Edgeeye: An edge service framework for real-time intelligent video analytics." Proceedings of the 1st international workshop on edge systems, analytics and networking. 2018.
* Surat Teerapittayanon, Bradley McDanel, and HT Kung. 2017. Distributed deep neural networks over the cloud, the edge and end devices. In Distributed Comput- ing Systems (ICDCS), 2017 IEEE 37th International Conference on. IEEE, 328–339.

---

> ### Author Response · Authors · 2022-08-02
> **Response to Reviewer HaF2**
>
> **Q1**: What do authors think about their possible way of defending such attacks through regularization? How good is that?
>
> **A1**: The current paper is on different attack methods and their evaluation. We plan to look at different defense techniques next. For starters, we evaluated a naïve method using L1 regularization selectively on the client-side model. The goal was to reduce the capability of the client-side model. Figure 4 (e) shows that L1 regularization can reduce extracted fidelity by 5 to 10%, suggesting this can be a possible defense technique. There could be other promising defense methods based on regularization. We plan to look at them in the near future.
>
> **Q2**: Have the authors also looked the LEAF datasets for the experiments? LEAF seems to be the standard dataset for FL papers, with natural client partitions and other FL constraints.
>
> **A2**: Thanks for bringing this up. We provide further empirical results on the official LEAF dataset FEMNIST (40K data, 62-class dataset) for the five attacks in Appendix - Table 8 using the same finetuning setting as in Table 2. The model is a VGG-11 model that achieves 74.62% validation accuracy.
>
> |    FEMNIST   | N | Craft |  GAN  |   GM  | Train | SoftTrain |
> |:------------:|:-:|:-----:|:-----:|:-----:|:-----:|:---------:|
> | Accuracy (%) | 2 | 53.20 | 10.59 | 56.67 | 70.32 |   75.70   |
> |              | 3 | 43.57 |  7.19 | 22.53 | 68.47 |   74.93   |
> |              | 4 | 43.04 |  5.27 |  9.87 | 68.80 |   74.42   |
> |              | 5 | 40.50 |  4.02 |  3.78 | 67.70 |   74.46   |
> | Fidelity (%) | 2 | 59.28 | 11.39 | 69.70 | 82.56 |   83.87   |
> |              | 3 | 48.10 |  7.35 | 25.14 | 77.52 |   81.24   |
> |              | 4 | 46.58 |  5.20 | 10.10 | 75.61 |   77.39   |
> |              | 5 | 42.11 |  3.80 |  3.68 | 71.97 |   76.30   |
>
> **Q3**: Some related work is missing. The authors mention that Split Learning is firstly introduced in reference "[Gupta and Raskar, 2018]" while the idea goes further back and there are already several papers on the field. For example (non exhaustive list): [Kang, Yiping et al. 2017], [Yousefpour, Ashkan et al. 2019], [Liu, Peng et al. 2018] and [Surat Teerapittayanon et al. 2017]
>
> **A3**: Thank you for pointing out this important issue. While the model split idea was proposed earlier, it was for distributed inference. This paper focuses on collaborative training and so we had quoted Gupta and Raskar's paper as it explicitly demonstrated the use of the model split area for such cases. We agree with your concern and have included these works in Section 2.1 to acknowledge them as the inventor of the “model split” idea. We maintain that [Gupta and Raskar, 2018] was the first to use the model split area for DNN training application.

---

> > ### Comment · Reviewer_HaF2 · 2022-08-08
> > **Thanks for the response**
> >
> > Hi,
> >
> > Thank you authors for addressing my concerns. I keep the score.

---

### Review · Ethics_Reviewer_SuLP · 2022-07-27

**Recommendation:**

Please write one more paragraph in Discussion or Conclusion to help readers to understand the ethical implications of the work. I believe that authors and reviewers agree about the ethical issues. The request is for a clearer explanation. This will be a service to the readers.

**Ethics Review:**

The request for ethics review comes from one reviewer, who would like the authors to add one more paragraph to clarify the societal impacts of the work. Three other reviewers did not see any ethical problems.

In my view, this is a reasonable request, and I believe that the authors can easily respond with the requested paragraph. The authors' motivation for this work is clearly ethical (as well as technical, of course). They already have the knowledge to write this paragraph. The requested paragraph will help to explain the authors' ethical motivation to a broader audience.

To state this clearly: There are NO ethical problems or issues with this paper. There is only a suggestion to explain the agreed ethical issues for a broader audience.

---

### Review · Ethics_Reviewer_JgGB · 2022-08-02

**Recommendation:** No remaining ethical issues.

**Ethical Issues:**

Yes

**Ethics Review:**

The original submission was lacking sufficient a discussion of societal impacts.

---

### Author Response · Authors · 2022-08-02
**Summary of changes in the updated version**

We thank all reviewers for their precious time and effort in putting in these reviews. We carefully considered the comments and revised our manuscript. In summary, we made the following changes in the revised version:

1. We revised the related work (Section 2.1) to include earlier papers on “model-split” idea.

2. We pointed out the similarity between Train-ME and the “model completion” method that was recently published in USENIX’ 22 in Section 4.2.

3. We ran experiments corresponding to a naive baseline, that directly uses the available training data to train the surrogate model, and updated Table 2 to include these results.

4. We added a “Broader Impact” subsection (Section 8) which states the societal impact of this work.

5. We included a diagram in Appendix A.2 to help better understand the process of all five proposed extraction attacks.

6. We included further empirical results of FEMNIST (one of LEAF datasets) in Appendix A.9.

7. We included time-cost measurement results for all five attacks in Appendix A.10.

---

### Meta-Review · Area_Chair_YGQc · 2022-08-27

**Recommendation:** Reject
**Confidence:** Less certain

**Metareview:**

The paper studies the vulnerability of split federated learning with model extraction attacks. The paper provides five attacks and evaluates them experimentally. The authors also provided additional experimental results during the author rebuttal.  While the topic and techniques are interesting, reviewers raise concerns about the novelty, and lack of experiments on standard FL datasets (e.g., LEAF) or large number of clients. While authors addressed some of these concerns during rebuttals, the paper can benefit from (a) explaining the novelty of the contributions (b) clarifying the assumptions made in the paper (c) explaining if the paper considers cross-device or cross-silo federated learning (b) adding more experiments on standard FL datasets and tasks.

**Award:**

No

---

### Decision · Program_Chairs · 2022-09-14

Reject